# Soluble Urokinase Plasminogen Activator Receptor (suPAR) in the Emergency Department (Ed): A Tool for the Assessment of Elderly Patients

**DOI:** 10.3390/jcm11123283

**Published:** 2022-06-08

**Authors:** Ria M. Holstein, Santeri Seppälä, Johanna Kaartinen, Mari Hongisto, Harri Hyppölä, Maaret Castrén

**Affiliations:** 1Faculty of Medicine, University of Helsinki, 00350 Helsinki, Finland; 2South Savo Social- and Healthcare District, Mikkeli Central Hospital, 50100 Mikkeli, Finland; santeri.seppala@essote.fi (S.S.); harri.hyppola@essote.fi (H.H.); 3Department of Emergency Medicine and Services, Faculty of Medicine, University of Helsinki, Haartmaninkatu 4, PL 340, 00029 HUS, 00350 Helsinki, Finland; johanna.kaartinen@hus.fi (J.K.); mari.hongisto@hus.fi (M.H.); maaret.castren@hus.fi (M.C.)

**Keywords:** Humans, Aged, Biomarkers, Receptors, Urokinase Plasminogen Activator, Prognosis, Emergency Service, Hospital, Risk Assessment, Patient Discharge

## Abstract

Emergency department (ED) overcrowding is a global issue setting challenges to all care providers. Elderly patients are frequent visitors of the ED and their risk stratification is demanding due to insufficient assessment methods. A prospective cohort study was conducted to determine the risk-predicting value of a prognostic biomarker, soluble urokinase plasminogen activator receptor (suPAR), in the ED, concentrating on elderly patients. SuPAR levels were determined as part of standard blood sampling of 1858 ED patients. The outcomes were assessed in the group of <75 years (=younger) and ≥75 years (=elderly). The elderly had higher median suPAR levels than the younger (5.4 ng/mL vs. 3.7 ng/mL, *p* < 0.001). Increasing suPAR levels were associated with higher probability for 30-day mortality and hospital admission in all age groups. SuPAR also predicted 30-day mortality when adjusted to other clinical factors. SuPAR acts successfully as a nonspecific risk predictor for 30-day mortality, independently and with other risk-assessment tools. Low suPAR levels predict positive outcomes and could be used in the discharging process. A cut-off value of 4 ng/mL could be used for all ED patients, 5 ng/mL being a potential alternative in elderly patients.

## 1. Introduction

Overcrowding of the emergency departments (EDs) is a widely discussed issue involving all EDs worldwide, caused by exit-blocks, decreasing numbers of ED beds and increasing need for acute care and, eventually, resulting in increased mortality rates, costs and prolonged length of stays (LOS) in the EDs. Consequently, this impairs the quality and safety of acute care [1,2,3,4].

The EDs face a rather heterogenous population of patients with both urgent and non-urgent medical conditions. Frail elderly patients are one of the most substantial and frequent visitors of the EDs, and their clinical presentation differs from the younger patient population: due to delayed, diminished or atypical clinical presentations and symptoms, the risk stratification of these patients is considered remarkably challenging. Additionally, due to age-related organ function declines, the patient population tend to have a higher risk for negative outcomes during their stay in the ED. For earlier mentioned reasons, the perspective of the aging population and consequential increases in elderly patients seeking care from the EDs is concerning [5,6].

The risk stratification in the EDs rely principally on vital-sign based track-and-trigger score systems, such as National Early Warning Score (NEWS) system. However, they can be insufficient in assessing the patients, especially elderly, with normal vital signs but with high risk of critical illness [7]. Therefore, tools for reflecting the underlying pathogenetic pathways of existing comorbidities as well as different acutellnesses are needed to improve the patient flow of the overcrowded EDs. The improved patient flow [8,9], would ideally result in more safe discharges and leave the hospital beds to the patients that require most clinical attention. Consequently, this could provide the EDs with increased resources and reduced costs, not to mention the advantages from the aspect of elderly hospitalization [10].

Prognostic biomarkers have been suggested as a potential tool for the clinical decision-making in the emergency setting [11]. One of the novel biomarkers, soluble urokinase plasminogen activator receptor (suPAR), is a nonspecific inflammatory biomarker, which is released in blood plasma when the urokinase plasminogen activator receptor (uPAR) is cleaved from the cell membrane of immunoactive cells such as monocytes, activated T-lymphocytes and macrophages in response to inflammatory stimuli. The plasma concentration of suPAR increases in both acute and chronic inflammatory states such as infectious diseases, sepsis, autoimmune diseases, malignancies, cardiovascular diseases and organ dysfunctions such as liver and kidney failure, when, in contrast, stays rather low in primary healthy individuals [12,13,14]. Furthermore, suPAR values in the general population increase with advancing age: a former study suggests that patients aged 74–89 years had significantly higher suPAR values than individuals between 24–66 years [15].

SuPAR has shown to have excellent prognostic value in both healthy individuals and in individuals with comorbidities [16,17,18,19]. In critically ill patients, suPAR levels are associated with increased risk of mortality, hospital admission, readmission rates as well as further complications [14,20,21,22,23]. Furthermore, suPAR values are studied to be strong predictors of mortality when adjusted with NEWS scoring, age and sex in ED patient population, and, interestingly, in hospitalized COVID-19 patients [24,25].

In contrast, low suPAR values have been observed to support the decision of discharge from the ED without increasing the risk for negative outcomes [26].

The EDs need additional tools for the risk assessment of their patients to improve their patient flow and avoid overcrowding. SuPAR is well understood when it comes to its characteristics and prognostic values. However, considering that aging increases suPAR levels, the optimal clinical setting for its use in the risk stratification of elderly ED patients is unclear. For that reason, in addition to evaluating the risk-predicting value of suPAR in the ED setting, this study aimed to determine the optimal cut-off values for the utilization of suPAR, concentrating on the elderly patient population.

## 2. Materials and Methods

### 2.1. Patient Population and Data Collection

This study was a prospective cohort study conceived in two Finnish hospital regions (Helsinki and Mikkeli). The included study population consists of unselected acute medical patients that sought care from the two study EDs between 4 March 2020 to 11 May 2020 (Mikkeli) or between 1 May 2020 to 31 May 2020 (Helsinki, Meilahti). The patient populations of the two hospitals were similar and consisted of patients from all medical specialties (internal medicine, surgery, trauma etc.).

The data were collected from the two hospital areas’ electronic health record systems (Uranus in Helsinki, Effica in Mikkeli). To be included in the study, the patient’s index admission was required to involve routine venous blood sampling and given consent (in Meilahti).

### 2.2. Biomarker Measurements

Plasma suPAR levels were incorporated as part of the standard blood sampling at the EDs. The actual measurement was carried out using suPARnostic^®^ Turbilatex assay (ViroGates A/S, Birkerød, Denmark) on a Cobas c501 clinical chemistry analyser (Roche Diagnostics Ltd., Espoo, Finland). The analyzing process was performed according to the manufacturer’s instructions. The other laboratory markers (C-reactive protein, creatinine, troponin T) were measured following regional standards. The suPAR values were available for the ED physicians in the same time frame as the other laboratory test results.

### 2.3. Statistics

The results are presented as numbers [N (%)] for categorical variables and as median [interquartile range (IQR)] for continuous variables. The patients were divided in two groups by age: (1) ≥75 years (=elderly) and (2) <75 years (=younger). For comparison of these groups, Fisher’s exact test or Pearson’s chi-squared test was used for categorical variables and Mann-Whitney U-test or Student’s *t*-test for continuous variables. Multivariable logistic regression analysis was used to determine independent risk factors for 30-day mortality, the results of which are presented as odds ratios (OR) with 95% confidence intervals (CI). We compared models with age group and suPAR interaction to ones without using likelihood-ratio tests (LRTs). Some unevenly and widely distributed values are presented on a logarithmic scale. NEWS scoring was excluded from the multivariable analysis due to missing data. A *p*-value less than 0.05 was considered statistically significant. The data was analyzed with SPSS Statistics Software 27.0 (IBM, Armonk, NY, USA).

### 2.4. Outcomes

The primary outcomes of this study were the all-cause mortality within 30 days of index admission and the number of discharges from the ED within 24 h of index admission. Secondary outcomes were hospital admissions, 7-day and 30-day readmissions and LOSs in the ED and in the hospital. All the outcomes were assessed in the whole population and separately in the elderly and in the younger.

## 3. Results

### 3.1. Whole Study Population and Age Groups

A total amount of 1858 (Mikkeli 1747 and Helsinki 111) patients were included in the study. Median age of the study population was 70 years (IQR 56–79) and 961 (52%) were women. 88 patients (5%) died within 30 days of index admission. Median length of stay (LOS) in the ED was 254 min (IQR 176–364), and 2 days (IQR 1–5) in the hospital.

The elderly constituted 36% (669/1858) of the patients with a female proportion of 48%. The rest 64% (1190/1858) of the patients were younger with a female proportion of 48%, respectively. The elderly had higher 30-day mortality compared with the younger (8% vs. 2%, *p* = 0.001)

The elderly were discharged from ED significantly less frequently during the first 24 h compared with the younger (30% vs. 54%, *p* < 0.001). Similar difference between the age groups was seen in hospital readmissions within 7 days of discharge (10% vs. 6%, *p* = 0.001). On the contrary, the amount of hospital admissions was higher in the elderly (68%) than in the younger (46%).

SuPAR values were available for 1845 (99.3%) patients. Median suPAR level in the whole study population was 4.1 ng/mL (IQR 3.3–6.0), 3.7 ng/mL (IQR 3.0–5.0) in the younger, and 5.4 ng/mL (IQR 4.1–7.7) in the elderly, respectively. Statistically significant differences between the age groups were additionally seen in the higher median glomerular filtration rates (GFRs) of the younger and in the higher median NEWS scores as well as median plasma levels of C-reactive protein (CRP) and troponine T (TnT) of the elderly.

For more detailed characteristics of the study groups, see Table 1.

### 3.2. SuPAR, Discharges and Mortality

The age groups were divided into three additional groups according to their suPAR levels: (1) suPAR <4 ng/mL, (2) suPAR 4–6 ng/mL, (3) suPAR >6 ng/mL. As Figure 1 shows, increasing suPAR levels were associated with decreasing proportion of discharged patients, the percentage of discharged patients within each suPAR category being: (1) 61% (2) 51% (3) 32% in the younger and (1) 45% (2) 34% (3) 19% in the elderly. Additionally, mortality rate increased along the suPAR level category: (1) 0.9% (2) 3% (3) 6% in the younger and (1) 0.7% (2) 6% (3) 15% in the elderly.

SuPAR levels were additionally observed between the discharged patients and the patients who died within 30 days of index admission. The differences were investigated in the two age groups. The median suPAR levels of the younger group who died within 30 days [5.8 ng/mL (IQR 4.1–9.7) were significantly higher than the levels of the younger discharged group [3.5 ng/mL (IQR 2.9–4.4)]. A similar trend was seen in the elderly group [7.6 ng/mL (IQR 5.5–10.1) vs. 4.5 ng/mL (IQR 3.7–5.8)]. Median suPAR values were higher in the elderly group, both in the discharged group and mortality group (See Figure 2).

### 3.3. Different SuPAR Cut-Offs in the ≥75 Years Group

To evaluate the predictive value of suPAR levels in the elderly population, the study’s outcomes were also assessed with different ranges using three separate suPAR cut-off values (0–4 ng/mL, 0–5 ng/mL and 0–6 ng/mL) in the elderly group separately (Table 2).

First, in the suPAR 0–4 ng/mL group, there were 153/23% elderly patients. In this group, 45% were discharged within 24 h, whereas 47% were admitted to hospital. One patient (0.6%) died within 30 days of index admission. The median LOS was 264 min (170–391) in the ED and 2 days (1.0–4.0) in the hospital.

Second, the suPAR 0–5 ng/mL group consisted of 289/43% elderly patients. In this group, 42% were discharged within 24 h of index admission and 57% admitted to hospital. Nine patients (3%) died within 30 days of index admission. The median LOS was 261 min (175–384) in the ED and 2 days (1.0–4.0) in the hospital.

Finally, in the suPAR 0–6 ng/mL group, there were 409/61% elderly patients, of which 23% were discharged within 24 h and 37% admitted to hospital. A total of 17 (4%) patients died within 30 days of index admission. The median LOS was 255 min (174–363) in the ED and 2.0 days (1.0–5.0) in the hospital.

### 3.4. Determination of Predictors for 30-Day Mortality—Unadjusted and Adjusted with Other Risk-Predicting Factors

The results for regression models can be found from Figure 3. SuPAR had an odds ratio (OR) of 1.23 (95% CI: 1.16–1.29) as a 30-day mortality predictor. When adjusting suPAR by age its OR slightly dropped: 1.18 (95%CI: 1.11–1.25). As age was correlated with suPAR, we kept it as a predictor and further adjusted the model with neurological and cardiovascular comorbidities, diabetes mellitus and logarithmized plasma levels of creatinine (krea) and troponine T (TnT). All of these had an association of equivalent level as when only adjusting suPAR with age. Only C-reactive protein (CRP) lowered the OR of suPAR considerably, when also adjusting with age the OR dropped to 1.09 (95%CI: 1.02–1.17). However, adding creatinine to the model with both age and CRP did not lower the OR of suPAR further (OR 1.09 95% CI: 1.01–1.17). Adding an interaction between the age and suPAR did not significantly increase fit of the model (LRT age as group *p* = 0.72, age as continous *p* = 0.63).

## 4. Discussion

The EDs are overcrowding and the current methods for the risk stratification are insufficient, especially in the elderly patient population. Thus, new methods for the assessment of these patients are needed. This study aimed to evaluate suPAR, a nonspecific prognostic biomarker, as a tool of this kind in the ED patient population. Additionally, the study analysed, for the first time according to our knowledge, the prognostic role and risk-predicting value of suPAR in the elderly population. According to LRT, adding interaction between suPAR and age did not improve any of the models significantly.

As a previous study working with the same research data has concluded [27], this study confirms that suPAR has prognostic value in predicting both negative and positive outcomes: patients with increased suPAR levels are more likely to die within 30 days of index admission, and patients with low suPAR levels are more likely to be discharged from the ED and survive within 30 days of index admission, regardless of age. Vice versa, the suPAR levels among patients who died within 30 days were significantly higher than the levels of the discharged patients. Additionally, our regression analysis indicates that suPAR acts as a predictor for 30-day mortality both independently and when adjusted with age, NEWS scoring, CRP and comorbidities such as diabetes mellitus, cardiovascular diseases and neurological diseases. When suPAR is simultaneously adjusted with three factors, the predictive value weakens (OR 1.09 (1.01–1.17).

Moreover, the study results suggest that suPAR levels are positively associated with age and the median suPAR level among the elderly population (5.4 ng/mL) is significantly higher than in the whole population (4.1 ng/mL) and in the younger population (3.7 ng/mL). Additionally, according to Figure 2, the median suPAR levels increase with age, regardless of whether the patient dies or is discharged from the ED.

However, despite the higher median suPAR levels, the study data suggests that the utilization of 6 ng/mL cut-off value would lead to excessive mortality rates in the elderly population (2.5%) and would thus impair the safety-related properties of low suPAR levels. (Table 2). The incidence of 30-day mortality was highest in the suPAR 0–6 ng/mL group when compared to the 0–4 ng/mL group and the 0–5 ng/mL group. Between the groups, an increase of this kind was additionally seen in both the number of discharges (8.0% increase from 0–4 ng/mL group to 0–5 ng/mL group, 5.1% increase from 0–5 ng/mL group to 0–6 ng/mL group) and the amount of 30-day readmissions (4.3% increase, 3.2% increase). The median length of stay in the ED or in the hospital did not significantly differ between the groups.

For that reason, a cut-off value of 4 ng/mL would successfully work as a predictor for both positive and negative outcomes in all patients, regardless of age. On the other hand, in the elderly, an elevation of the cut-off value from 4 ng/mL to 5 ng/mL resulted in a significant increase in the proportion of discharges (10.3% vs. 18.3%) but only one death within 30 days of index admission.

SuPAR is a nonspecific biomarker, and elevated suPAR values can be caused by chronic non-acute as well as acute diseases. The aim of this study was to determine if suPAR can predict negative outcomes in an unselected patient population with various chronic illnesses, especially in the elderly. According to the study results, suPAR predicts mortality in this group, regardless of age. However, due to its unspecificity, suPAR is not a diagnostic tool. For that reason, suPAR should be used more as a directional prognostic tool alongside other clinical features and assessment methods such as clinical examination, scoring systems and other laboratory markers. Judging by previous study and the data presented in this manuscript, suPAR could thus be used in the decision to either admit or discharge the ED patient.

### Limitations

As with the majority of studies, this study is subject to limitations. First, the ED physicians were conscious of the patients’ suPAR results in Mikkeli but not in Helsinki, and therefore the evaluation of the effects on the outcomes is not possible. Second, the smoking habits of the included patients were not taken into account, regardless of knowing that regular smokers have approximately 1 ng/mL higher suPAR levels than non-smokers [28,29]. Third, as drawn blood samples and given consent in Meilahti were required for the inclusion, the study excluded the patients with minor clinical issues, mental issues or nurse visits, for example. Additionally, the patients that were not able to give a consent in Melahti were excluded from the study.

## 5. Conclusions

The study results suggest that suPAR levels were clearly elevated in the ED patients, the elderly patients displaying the highest levels. However, age and suPAR were not associated with 30-day mortality. High suPAR concentrations were associated with higher mortality and lesser probability to be discharged from the ED. Furthermore, as a nonspecific prognostic biomarker utilized in the ED, suPAR successfully predicts all-cause 30-day mortality in all age groups. SuPAR maintains its predictive value when it is used with other commonly used risk assessment tools. Low suPAR values can work as a support in discharging patients from the ED without increasing the risk of negative outcomes.

For all the patients arriving at the ED, the safest cut-off value for suPAR would be 4 ng/mL. On the other hand, a cut-off value of 5 ng/mL should be considered as a potential alternative in the elderly population. The cut-off value of 6 ng/mL should not be utilized.

Our study confirmed that suPAR could successfully act as an addition to the risk assessment of elderly patients and the patients of which the current risk stratification methods fail to identify, especially as these patients are one of the most time- and resource-consuming patients of the ED.

## Figures and Tables

**Figure 1 jcm-11-03283-f001:**
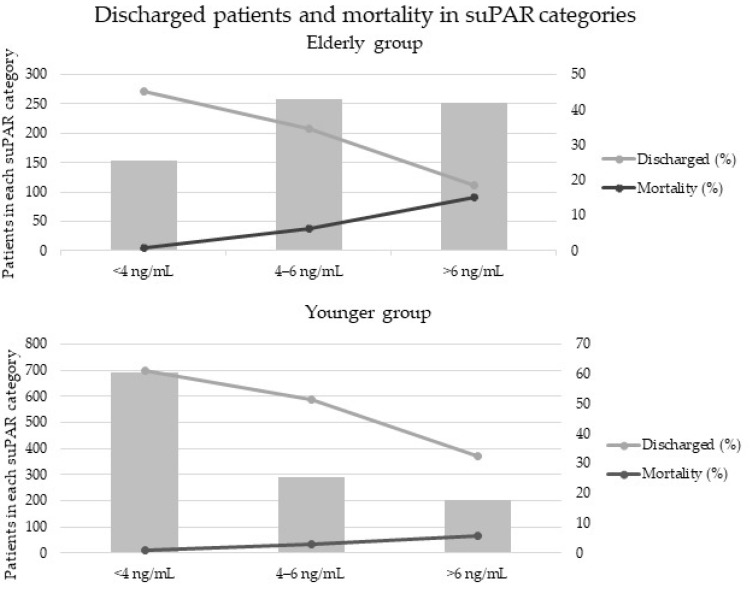
Discharged patients and mortality in different suPAR categories and age groups.

**Figure 2 jcm-11-03283-f002:**
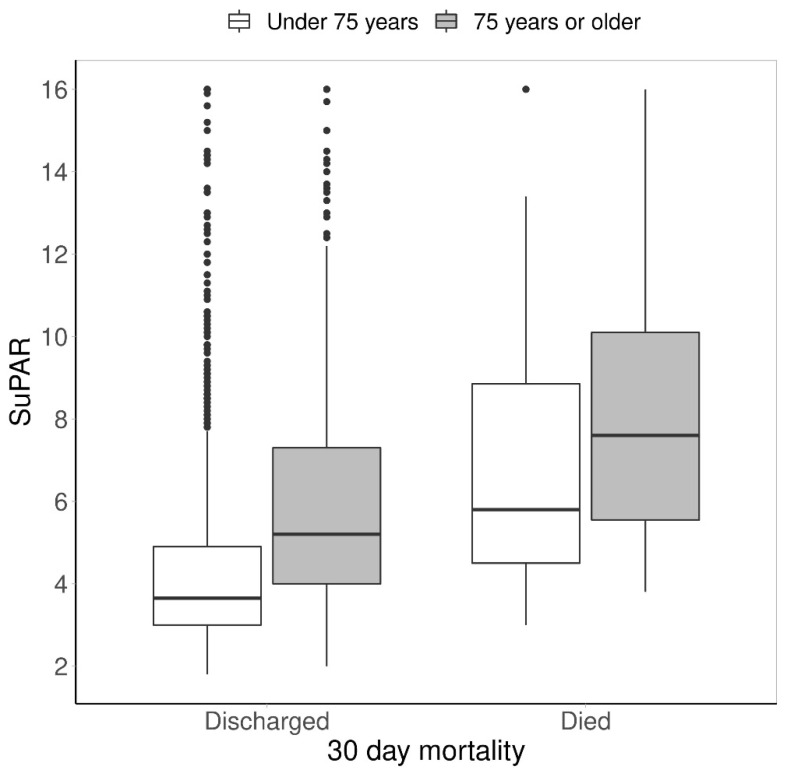
Box Plot of the association between suPAR levels, age, discharge and 30-day mortality.

**Figure 3 jcm-11-03283-f003:**
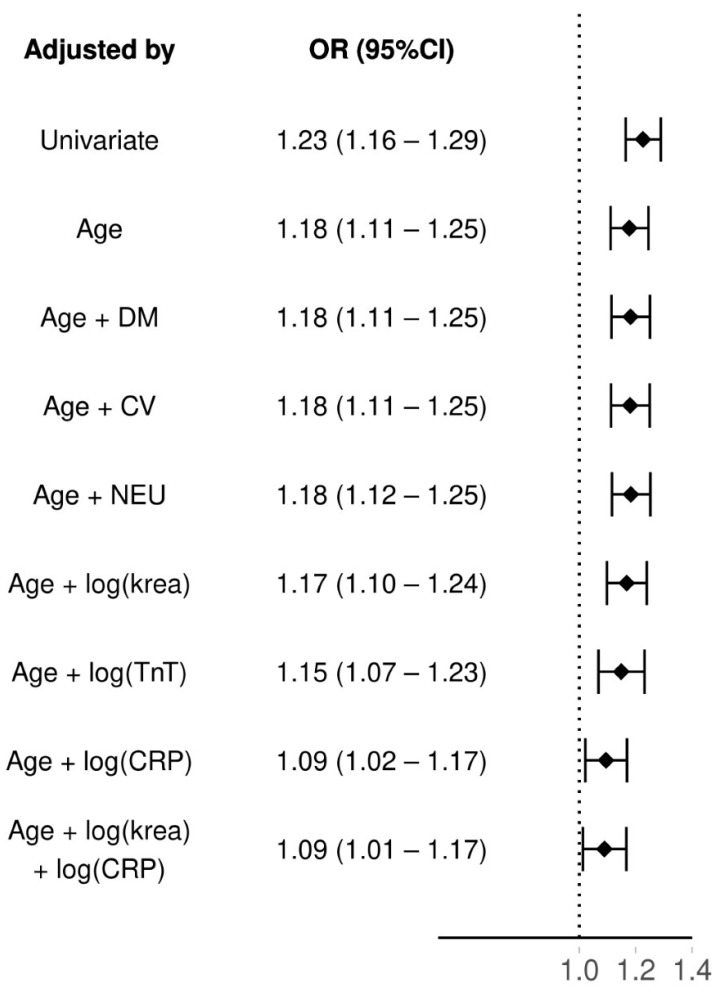
Multivariable analyses of suPAR and 30-day mortality adjusted with age and other clinical factors. suPAR = soluble urokinase plasminogen activator receptor, OR = odds ratio, CI = confidence interval, DM = diabetes mellitus, CV = cardiovascular disease, NEU = neurological disease, NEWS = National Early Warning Score, log(x) = the outcome x on a logarithmic scale, krea = plasma creatinine, TnT = troponin T, CRP = C-reactive protein.

**Table 1 jcm-11-03283-t001:** Characteristics of the study population—whole population and age groups.

*Variables*	*All*	*<75 Years*	*≥75 Years*	*p-Value*
*N (%)*	1858 (100)	1190 (64.0)	668 (36.0)	
*Female sex (N, %)*	961 (52)	574 (48)	358 (58)	<0.001
*Age, median (IQR)*	70 (56–79)	61 (45–70)	83 (79–88)	<0.001
*suPAR ng/mL, median (IQR)*	4.1 (3.3–6.0)	3.7 (3.0–5.0)	5.4 (4.1–7.7)	<0.001
*Creatinine, median (IQR)*	76 (63–96)	72 (61–87)	87 (70–115)	<0.001
*GFR, mL/min, median (IQR)*	80 (60–95)	90 (75–103)	61 (44–77)	<0.001
*GFR > 60 mL/min, N (%)*	1374 (74.0)	1028 (86.4)	346 (51.8)	
*GFR 30–60 mL/min, N (%)*	359 (19)	113 (10)	246 (37)	
*GFR < 30 mL/min, N (%)*	94 (5.1)	21 (1.8)	73 (10.9)	
*CRP, median (IQR)*	3 (3–19)	3 (3–16)	4 (3–22)	0.012
*Leukocytes, median (IQR)*	7.6 (6.0–9.8)	7.6 (6.0–9.7)	7.5 (6.0–10.1)	0.417
*TnT, median (IQR)*	15 (8–29)	10 (6–17)	25 (16–45)	<0.001
*NEWS score, median (IQR)*	1 (0–2)	0 (0–2)	3 (1–5)	<0.001
*30D pre-ER, N (%)*	379 (20.4)	240 (20.2)	126 (20.5)	0.951
*Discharge < 24 h, N (%)*	847 (45.6)	639 (53.7)	187 (30.4)	<0.001
*Readmission 7 D, N (%)*	166 (8.9)	124 (10.4)	36 (5.9)	0.001
*Readmission 30 D, N (%)*	382 (20.6)	250 (21.0)	123 (20.0)	0.668
*Mortality 30 D, N (%)*	82 (4.4)	27 (2.3)	51 (8.3)	0.001
*Admission to hospital*	1001 (53.9)	552 (46.4)	417 (67.8)	0.008
*LOS, minutes, median (IQR)*	254 (176–364)	250 (171–364)	258 (184–366)	0.074
*Hospital stay, days, median (IQR)*	2.0 (1.0–5.0)	1.0 (1.0–4.0)	3.0 (1.0–7.0)	<0.001

IQR = interquartile range, suPAR = soluble urokinase plasminogen activator receptor, GFR = glomerular filtration rate, NEWS = National Early Warning Score, LOS = length of stay, 30D pre-ER = number of patients who sought care from the ED 30 days before the study’s index admission.

**Table 2 jcm-11-03283-t002:** Different suPAR cut-off values in the ≥75 years group.

*Variables*	*≥75 y*	*≥75 y & suPAR 0–4 ng/mL*	*≥75 y & suPAR 0–5 ng/mL*	*≥75 y & suPAR 0–6 ng/mL*
*N (% of ≥75 y)*	668	153 (22.9)	289 (43.3)	409 (61.2)
*Female sex, N (% of ≥75 y)*	387	95 (14.2)	169 (25.1)	239 (35.8)
*Age, years, median (IQR)*	82	79 (77–83)	80 (77–84)	81 (78–86)
*suPAR ng/mL, median (IQR)*	5.4 (4.1–7.6)	3.5 (3.2–3.7)	3.9 (3.5–4.4)	4.3 (3.7–5.1)
*Creatinine, median (IQR)*	87 (68–114)	74 (63–88)	77 (65–93)	79 (66–97)
*GFR, mL/min, median (IQR)*	61 (44–78)	75 (62–83)	70 (58–82)	68 (54–81)
*GFR > 60 mL/min mL/min, N (%)*	346	118 (71.5)	210 (70.9)	265 (64.3)
*GFR 30–60 mL/min, N (%)*	246	35 (21.2)	75 (25.3)	128 (31.1)
*GFR <30 mL/min, N (%)*	73	No patients	2 (0.7)	13 (3.2)
*CRP, median (IQR)*	4 (3–23)	3 (3–3)	3 (3–4)	3 (3–6)
*Leukocytes, median (IQR)*	7.5 (6.0–10.1)	6.8 (5.9–8.3)	6.9 (5.9–8.7)	7.1 (5.9–8.8)
*TnT, median (IQR)*	24 (15–44)	15 (10–23)	17 (12–25)	19 (13–30)
*NEWS score, median (IQR)*	3 (1–5)	0 (0–1)	0 (0–2)	1 (0–2)
*30D pre-ER, N (% of ≥75 y)*	139	27 (4.0)	51 (7.6)	78 (11.7)
*Discharge <24 h, N (% of ≥75 y)*	208	69 (10.3)	122 (18.3)	156 (23.4)
*Readmission 7 D, N (% of ≥75 y)*	42	12 (1.8)	24 (3.6)	32 (4.8)
*Readmission 30 D, N (% of ≥75 y)*	132	36 (5.4)	65 (9.7)	86 (12.9)
*Mortality 30 D, N (% of ≥75 y)*	55	1 (0.15)	9 (1.3)	17 (2.5)
*Admission to hospital (% of ≥75 y)*	449	72 (10.8)	165 (24.7)	250 (37.4)
*LOS, minutes, median (IQR)*	258 (183–366)	264 (170–391)	261 (175–384)	255 (174–363)
*Hospital stay, days, median (IQR)*	3.0 (1.0–7.0)	2 (1.0–4.0)	2 (1.0–4.0)	2.0 (1.0–5.0)

IQR = interquartile range, GFR = glomerular filtration rate, NEWS = National Early Warning Score, LOS = Length of stay, 30D pre-ER = number of patients who sought care from the ED 30 days before the study’s index admission.

## Data Availability

The data presented in this study are available on request from the corresponding authors.

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
