# Peer review of "Soluble Urokinase Plasminogen Activator Receptor (suPAR) in the Emergency Department (Ed): A Tool for the Assessment of Elderly Patients"

_jcm, 2022, doi:10.3390/jcm11123283_

Round 1
Reviewer 1 Report
The authors present a 2-center cohort study including the heterogenous groups of ED patients / acute medical patients. They examine the association between the levels of a prognostic biomarker, suPAR (exposure), and 30-day mortality (outcome) and study the interaction with age-group (<75 vs. >75y). SuPAR is already known for its predicitve performance for mortality and severe illness e.g. in acute pancreatitis (doi: 10.1002/jcla.23097) and in ICU patients (https://doi.org/10.1038/s41598-021-96352-1) to only mention two studies. A big cohort study (n>17,000) of ED patients was performed in 2018 (DOI: 10.1097/CCM.0000000000003441). A recently published protocol for a systematic review and meta-analyis (10.1136/bmjopen-2019-036125) is currently ongoing.
In my opinion, the authors have a good dataset and the manuscript is nicely written. The results were more or less already known. I suggest a more sophisticated statistical analysis to further enhance the quality of the manuscript.
Introduction/Abstract:
- For me the relationship of a prognostic biomarker and overcrowding is not very straight forward. Maybe the authors
Methods:
- Please elaborate on the "acute medical patients", what exactly are these patients? And how are they determined?
- How were missing data treated?
- I suggest spliting the cohort into a training and a validation set
- If I understand the idea correctly, predicting factors in elderly and younger patients are identified separately. That said, I do not understand why comparing the cohorts regarding age make sense
- Consider presenting one model with an interaction of suPAR levels and age group
- Consider presenting a Cox Regression additionally or instead to the logistic regression analysis.
- Consider presenting more model characeristics for the multivariable model (see here for the 10 relevant criteria: doi: 10.1016/j.jclinepi.2003.05.003)
- Consider adding ICU admission to the outcomes
- Why were patients in Mikkeli only enrolled for one week? This is somehow odd
- What is the rational to choose the cut-off of 75years?
Results:
- As said the comparison of and description of young vs. old comparison is somewhat not very informative
- Consider presenting Box Plot Digrams of suPAR in different age groups (maybe more than the two age groups with the arbitraty cut off of 75) and stratified by 30day mortality to see the magnitude of the differences
- I wonder if stacked bar charts would be clearer than the actual figure 1
- Figure 2: consider changing the name to multivariaBle througout the manuscript (see doi: 10.2105/AJPH.2012.300897), the figure has poor quality, I do not understand why the for the NEWS score it is not adjusted, furthermore it is not vlear to me why you adjust only for one factor except for the last line (why CRP and Crea and age? and not NEWS, CV or DM) or is the effect size adjusted on all of the above?
- sometimes the Font seems to be different e.g. 187-189
- I still think that the rational for this study, is the analysis of the interaction of SuPAR levels and age. Consider using techniques for visualisationas suggested here (https://pubmed.ncbi.nlm.nih.gov/22652348/ or https://stats.oarc.ucla.edu/stata/faq/how-can-i-explain-a-continuous-by-continuous-interaction-stata-12/#:~:text=First%20off%2C%20let%27s%20start%20with,models%20often%20contain%20interaction%20terms etc.
Discussion:
- Use introduced abbreviations throughout the manuscript (line 195: emergency department <=> ED)
- The discussion is very short especially considering the number of previous studies
Author Response
Dear Reviewer 1,
Thank you for the informative and insightful comments and suggestions. Please see attached the point-by-point response file to the Editor and the all Reviewers.

Reviewer 2 Report
Good morning Authors:
1.- In Table 2.
a.- You talk about: suPAR <4ng/ml, suPAR <5ng/ml, suPAR <6ng/ml.
But, Sensu stricto, <6ng/ml include ; 5,4,3,2,1,0 etc
and <5ng/ml include; 4,3,2,1,0 etc.
and <4ng/ml include; 3,2,1,0 etc.
Don´t you think it´s better to write a range?
Ex: 0 - 4, 4.1 - 5, 5.1 - 6 ??
b.- Table 2
Hospital Days. Median 2.
In those with suPAR <6ng/ml the maximum (Hospital Days) was 5 days, It looks short.
Were not the patients with serious diseases?
2.- In Figure 2.-
In Multivariate analysis
a.- All Data, Adjusted by log (Krea) + log (CRP) + Age
O.R= 1.09 ( 95% IC) (1.00 - 1.18)
It´s OR near 1 and in IC is included 1, so, no good precision.
b.- >= 75 years.
Adjusted by log (krea) + log (CRP) + Age
O.R = 1.1 95% IC ( 1.00- 1.20)
O.R near 1, and included 1 in CI, so, no good precision.
Do you have any comment?
3.- I recommend to read these articles:
a.- Atul Kakar, Pooja Rani, Tawi Batra, Riswana Hassan , Sangeeta Choudhury
A comparative analysis of serial measurements of soluble Urokinase type Plasminogen Activator Receptor (suPAR) and C Reactive Protein in patients with moderate COVID-19 a single center study from India.
MedRXiv , BMJ Yale, March, 30, 2022.
b.- Tobias P. Schmidt, Walid Al-banna, Miriam Weiss, Michael Veldeman et al
The role of soluble Urokinase Plasminogen Activator Receptor (suPAR) in the context of Aneurysmal Subarachnoid Hemorrhage (ASAH) a Prospective Observational Study.
Frontiers in Neurology Stroke.
Front. Neurol, March 10, 2021.
Author Response
Dear Reviewer 2,
Thank you for your insightful and informative comments and suggestions. Please find attached the point-by-point response file that replies to all reviewers' comments.

Reviewer 3 Report
This is a well-designed cohort study with well control/adjustment that shows suPAR could be used as an addition to the risk assessment for the patients whether to stay or be discharged. For example, a cut-off value of 5 ng/ml can be considered an indicator for the elderly population to stay in or be discharged from ED. The suPAR also could be used to successfully predict all-cause 30-day mortality in all age groups. These findings are truly helpful in ED in a practical manner. Some comments are as follows: (1) If possible, please show more studies relevant to the clinical use/findings of suPAR in the introduction. For example, older individuals tend to have higher suPAR levels than the younger; or low suPAR values have been observed to support the decision of discharge from the ED without increasing the risk for negative outcomes. (2) If possible, please have more other references to be compared with your findings in the discussion. (3) Please make sure the fonts are all the same in the content, e.g., fonts in Line 160, 163, etc... (4) In Figure 1, please make the legends and axis clear: Are the lines or bars in the figure should be referred to on the right or left axis? (5) In Table 2, please remove the term "of ≥75y" in the parentheses, and just keep the "%" in the parentheses because "of ≥75y" looks redundant. For example, remove the "of ≥75y" in "Discharge < 24h, N (% of ≥75y)" which remains "Discharge < 24h, N (%)" (6) Try using "Number (%)", e.g., 153 (23%) in Line 161, rather than 153/23%, to be more readable. (7) In Line 187, it mentioned Figure 3, but I don't see any Figure 3 in the content. Please provide the figure or clarify if it should be some other figure.Author Response
Dear Reviewer 3,
Thank you for your informative and insightful comments and suggestions. Please see attached the point-by-point response file answering to all of the reviewers' comments.

Reviewer 4 Report
Soluble Urokinase Plasminogen Activator Receptor (suPAR) in the Emergency Department (ED): a tool for the assessment of elderly patients
Summary: This study is a prospective cohort study that measured the concentration of soluble urokinase plasminogen activator receptor in the Emergency Department of two Finnish hospitals regions. The study aimed to determine the optimal cut-off values for the utilization of suPAR concentrating on the elderly patient population.
The study has merit, but I think the results are not clearly presented and I am not convinced by the results that the biomarker is helpful. The authors achieved their objective of determining the cut-off values but these values may be skewed by comments below.
Critique:
Introduction: page 3 line 80: change rise to increase.
Material & Methods: Was patient consent necessary? Were all patients who presented to the Emergency Department included in the study or only those that had blood drawn?
Results and Discussion: There is a large discrepancy between the patient numbers between the two hospitals; was that taken into account in the analysis? Does one hospital see more elderly patients than the other?
The analysis is complicated so I think Figure 1 could be presented in additional figures. I found it difficult to follow.
Regarding the >75-year-old group, I find this confusing. <4 group had 153 patients with 47% admitted; <5 group (does this imply levels between 4 and 5?) had 289 patients with 57% admitted and the <6 group (does this imply levels between 5 and 6?) had 409 patients with 37%. The total is 851 patients, however, in table 1 the total number of patients >75 was 668? Unless I am reading this wrong, this does not add up correctly. Grossly interpreting this data, it appears the >6 group had the lowest admission rate.
What is the value of suPAR in the general population broken down by age?
suPAR is elevated in cancer patients, was that taken into account in the analysis? A large number of cancer patients, especially in the elderly, could skew your results.
Was suPAR increased in patients that had acute cardiovascular disease, cardiac (myocardial infarction) or brain (stroke)?
Inspection of the data suggests using GFR as an indicator of morbidity or mortality. You might want to analyze this data point.
Is the ultimate goal of this biomarker to be used in decision to admit or discharge?
Author Response
Comments and Suggestions for Authors
Soluble Urokinase Plasminogen Activator Receptor (suPAR) in the Emergency Department (ED): a tool for the assessment of elderly patients
Summary: This study is a prospective cohort study that measured the concentration of soluble urokinase plasminogen activator receptor in the Emergency Department of two Finnish hospitals regions. The study aimed to determine the optimal cut-off values for the utilization of suPAR concentrating on the elderly patient population.
The study has merit, but I think the results are not clearly presented and I am not convinced by the results that the biomarker is helpful. The authors achieved their objective of determining the cut-off values but these values may be skewed by comments below.
Author response: We thank the third reviewer for the insightful comments and suggestions! Please see below the point-by-point responses to all the comments. All the changes to the updated manuscript have been written in the responses in addition to the MS Word file.
Critique:
Introduction: page 3 line 80: change rise to increase.
Author response: Thank you for the correction. The mistaken use of ‘rise’ instead of ‘increase’ has now been corrected on Line 95.
Material & Methods:
Was patient consent necessary?
Author response: We thank the reviewer for the great question. Patient consent was required for the inclusion in Meilahti. In Mikkeli, consent was not required.
We think that patient consent in Meilahti was, indeed, not necessary, and could have caused a bias as the most critically ill patients could not have been able to give a consent. To add, only the patients who had blood samples taken were included in the study. This can also have excluded patients that had only minor clinical issues or clinical issues not requiring blood samples. However, we think that this risk of bias has not been a significant issue, as we know that primary healthy people tend to have lower suPAR levels, and, additionally, we aimed to evaluate suPAR as a risk predictor of critical severe illness and mortality. To add, the amount of patients excluded due to earlier mentioned reasons (the patients without given consent in Meilahti or drawn blood samples in both EDs) has been quite small, and to our knowledge, has not significantly distorted the interpretation of our results.
Finally, the Ethical Committee of Helsinki University Hospital decided that given consent was required to include a patient to the study in Meilahti ED, and, thus, informed consent was obtained from the included patients.
We fully agree that this is good to be mentioned in the manuscript. Additionally, the inclusion criteria were incorrect in the submitted manuscript. Therefore, we have corrected the issue and added a phrase in the Limitations sections that clarifies the takes the earlier mentioned risk of bias into account (can be found on Lines 313-316)
Were all patients who presented to the Emergency Department included in the study or only those that had blood drawn?
Author response: As mentioned above, a patient was included in the study if its ED visit required blood sampling regardless of the study. This is also mentioned in the Materials & Methods section (Line 112). The possible risk of bias derived from the inclusion criteria has been discussed above, and, to add, has now taken into account, by adding a phrase concerning this issue to the Limitations section (Lines 313-316).
Results and Discussion: There is a large discrepancy between the patient numbers between the two hospitals; was that taken into account in the analysis? Does one hospital see more elderly patients than the other?
Author response: We thank the reviewer for the great question. The patient populations between the two hospitals did not differ significantly, which enabled the fusion of the two study data. To add, the discrepancy between the patient numbers is most likely caused by the longer study period in Mikkeli (the 1-month study period of Meilahti cut in half due to the ongoing pandemic). To add, as mentioned in the previous response, a given consent was obtained from the included patients in Meilahti. Patient consent was not required from the patients in Mikkeli, so more patients could be included in a same period of time.
Thank you for bringing this up. The issue has not been mentioned in the manuscript, so we added more information to the “Materials and methods” section of the manuscript for clarity (Lines 107-109).
The analysis is complicated so I think Figure 1 could be presented in additional figures. I found it difficult to follow.
Author response: We would like to thank the reviewer for this suggestion. Figure 1 was indeed difficult to follow, as both Reviewer 1 and Reviewer 3 mentioned. Therefore, we created a new Figure 1 that consists of two separate bar-line-graphs (elderly and younger) instead of one graph. We hope that the new Figure 1 is more easy to read. At first our plan was to create a stacked bar chart but found it similarly difficult to follow due to the great amount of variables and axises, and therefore, decided to split the graph in two parts.
Regarding the >75-year-old group, I find this confusing. <4 group had 153 patients with 47% admitted; <5 group (does this imply levels between 4 and 5?) had 289 patients with 57% admitted and the <6 group (does this imply levels between 5 and 6?) had 409 patients with 37%. The total is 851 patients, however, in table 1 the total number of patients >75 was 668? Unless I am reading this wrong, this does not add up correctly. Grossly interpreting this data, it appears the >6 group had the lowest admission rate.
Author response: Thank you for the comment. The results of the evaluation of different suPAR cut-off-values in the >75-year-old group have indeed been incoherently reported. The <4 ng/mL group is the group of >75-year-olds that had a suPAR value between 0-4 ng/mL and the <5 ng/mL group is the group consisting >75-year-olds with a suPAR value between 0-5 ng/mL and etc. The amount of patients therefore increases as the “maximum” threshold suPAR value (“cut-off value) increases. In other words, the patients in the <4 ng/mL group are also included in the <6 ng/mL group as their suPAR value is lower than 6 ng/mL.
The incoherent reporting of the results in Table 2 was also mentioned by Reviewer 2, and we fully agree with your observations. However, we think that the analysis of the cut-off-values in the elderly group is otherwise informative. For that reason, we changed the suPAR values of the groups to ranges (the <4 ng/mL group to 0-4 ng/mL group and <5 ng/mL group to 0-5 ng/mL group etc.) The changes can be found in the Results and Discussion sections and Table 2. We hope that the corrections are clarifying enough.
What is the value of suPAR in the general population broken down by age?
Author response: To cite two articles discussing this topic (found on our References list), the suPAR values of general primary healthy population tend to be lower than the diseased population. Furthermore, the suPAR values of older individuals tend to be higher than of younger individuals (partly resulting from the differences in the amounts of the morbidities). For example, according to the 1) article), the median (IQR) suPAR values of Caucasian general population were 3.16 (2.65–3.51) in the population of younger individuals [median (IQR) age being 46 (36.60)] and 3.79 (3.17–4.52) in the population of older individuals [median (IQR) age being 79 (77-82)]. The 2) study also confirms that lifestyle habits affect suPAR levels in. a general population cohort [mean (SD) age being 46.4 (7.8) and median (IQR) suPAR being 3.26 (1.41)].
- Wlazel, R.N., Szwabe, K., Guligowska, A.et al. Soluble urokinase plasminogen activator receptor level in individuals of advanced age. Sci Rep 10, 15462 (2020).
- Haupt, T. H., Rasmussen, L., Kallemose, T., Ladelund, S., Andersen, O., Pisinger, C., & Eugen-Olsen, J. (2019). Healthy lifestyles reduce suPAR and mortality in a Danish general population study.Immunity & ageing : I & A, 16, 1. https://doi.org/10.1186/s12979-018-0141-8
To clarify the characteristics of suPAR in the general population, a phrase concerning this topic has been added to the Introduction section (Lines 77-78).
suPAR is elevated in cancer patients, was that taken into account in the analysis? A large number of cancer patients, especially in the elderly, could skew your results.
Author response: We want to thank the reviewer for bringing this important topic up. SuPAR values are indeed elevated in patients with malignancies (Line 82 in manuscript) as well as many other chronic diseases, and the underlying diseases can truly affect the study results. That said, we agree with the reviewer: the potential confounding factors (the existing comorbidities in this case) should be precisely determined and the potential risk of bias evaluated.
It is certainly correct that the suPAR values should not be interpreted without seeing the whole clinical picture. Thus, suPAR should be used more like other commonly used laboratory markers (C-reactive protein, TnI, creatinine…) by understanding the different pathophysiologies behind the elevated values. Additionally, ED physician should be conscious of the marker’s clinical “weaknesses” and the possible confounding factors increasing or decreasing the value.
However, a significant portion of the malignancies are diagnosed in the ED or during the incoming hospital period. In other words, all patients with cancer can’t be identified during the ED visit. That is why the exclusion of patients with underlying malignancies would have led to a significantly high risk of bias. Also, further analysis on the underlying comorbidities of a study population of this size would have been very difficult.
Cancer is also a standalone risk factor in the ED. For example, patients with cancer are of risk of higher mortality, when presented to the ED with sepsis and bactaraemia, when compared to non-cancer patients. (Abou Dagher G, El Khuri C, Chehadeh AA, Chami A, Bachir R, Zebian D, Bou Chebl R. Are patients with cancer with sepsis and bacteraemia at a higher risk of mortality? A retrospective chart review of patients presenting to a tertiary care centre in Lebanon. BMJ Open. 2017 Mar ) Thus elevated suPAR values in cancer patients are representative of the mortality risk of the patient group.
We evaluated suPAR in an unselected population of patients with many chronic diseases. The aim of this study was to determine if the biomarker could predict negative outcomes in a heterogenous and multi-diseased population of this kind. In other words, the aim of this study was not to investigate how suPAR acts in specific diseases (as it has been already largely studied).
Since this is a certainly important topic to discuss, we added a whole paragraph in the Discussion section (Lines 299-307).
Was suPAR increased in patients that had acute cardiovascular disease, cardiac (myocardial infarction) or brain (stroke)?
Author response: Thank you for the important question. SuPAR values have studied to be elevated in patients with cardiac disease such as acute coronary syndrome (ACS) and congestive heart failure. SuPAR elevation can be an independent predictor of mortality in these medical conditions. 1) (found on the manuscripts References list).
We strongly agree on the reviewer’s point of view - the evaluation of suPAR’s properties in specific comorbidities is an extremely important field of research and should absolutely be studied more in the future. However, as discussed in the response above, we think that the further investigation of suPAR values in specific comorbidities did not compliment this particular study’s aims and aspects (as we wanted to evaluate suPAR in an unselected and multi-diseased group of patients). To add, the addition of an analysis of this kind would have resulted in a rather wide set of results.
- Velissaris D, Zareifopoulos N, Koniari I, Karamouzos V, Bousis D, Gerakaris A, et al. Soluble Urokinase Plasminogen Activator Receptor as a Diagnostic and Prognostic Biomarker in Cardiac Disease. Journal of Clinical Medicine Research. 2021 Mar 1;13(3):133–42.
Inspection of the data suggests using GFR as an indicator of morbidity or mortality. You might want to analyze this data point.
Author response: Thank you for the suggestion. GFR can indeed act as a predictor for mortality. In our study, instead of GFR, we used the creatinine value in the further analysis. That is due to multiple reasons: 1) in Finland, the plasma creatinine is more used and 2) creatinine is widely understood and 3) GFR is not always automatically calculated in the ED.
Is the ultimate goal of this biomarker to be used in decision to admit or discharge?
Author response: Thank you for the question, you are indeed correct on both points. We think, judging by the data presented in this manuscript, that low suPAR values can be used as a support for the decision to discharge, and high values can be used as a support for the decision to admit the patient. The results of our study and previous studies indicate that suPAR could be used in this context. Thus, with the help of the reviewer’s excellent question, we added a phrase concerning this idea in the Discussion section (Lines 299-307).

Round 2
Reviewer 1 Report
The authors put effort to increase the quality of their manuscript and replied to all of my comments satisfactory.
However, due to a 10-days-deadline, the authors could not perform the required statistical analysis to further increase the quality of the manuscript.
Thus, I suggest, that the authors will be given the time they need by the editor, so that the analysis could be further impoved. I recommend that the analysis will be supervised by their statistician Mitja Lääperi. Furthermore, the statistician Mitja Lääperi should be consulted if it makes sense to present the impact of SUPAR on mortality stratified by the two age group if there is no significant interaction found in a multivariable model.
I am looking forward to the revised version of the manuscript including the more sophisticated statistical analysis.
Author Response
Dear Reviewer 1,
We are truly thankful for your insightful comments. Please find attached a point-by-point response file to your comments. All the new responses are marked in green colour.
Best regards,
Ria Holstein and Santeri Seppälä

Reviewer 4 Report
The revised manuscript is significantly improved . The additional figures are helpful as well as the added text. The line numbers mentioned in the response to reviewers do not match, for example, the authors refer to lines 107-109 but lines 96-114 are missing. Maybe this is just a editing problem as inspection of the revised manuscript seems more coherent. In addition, I am still a bit unclear concerning the patient count regarding the number of patients in the >75 year old group, but I will accept their explanation.
Author Response
Dear Reviewer 3,
We are truly thankful for your insightful comments. Please find attached our point-by-point response file with new responses marked in green color. Please also find attached the updated manuscript.
